# That was the last straw, we need more:
## Are Translation Systems Sensitive to Disambiguating Context?

**Jaechan Lee**[♡]   **Alisa Liu**[♡]   **Orevaoghene Ahia**[♡]   **Hila Gonen**[♡]   **Noah A. Smith**[♡♣]

[♡]Paul G. Allen School of Computer Science & Engineering, University of Washington
[♣]Allen Institute for AI

{chan0369,alisaliu,oahia,nasmith}@cs.washington.edu,hilagnn@gmail.com

## Abstract

The translation of ambiguous text presents a challenge for translation systems, as it requires using the surrounding context to disambiguate the intended meaning as much as possible. While prior work has studied ambiguities that result from different *grammatical* features of the source and target language, we study semantic ambiguities that exist in the source (English in this work) itself. In particular, we focus on idioms that are open to both literal and figurative interpretations (e.g., *goose egg*), and collect TIDE,[1] a dataset of 512 pairs of English sentences containing idioms with disambiguating context such that one is literal (*it laid a goose egg*) and another is figurative (*they scored a goose egg*, as in a score of zero). In experiments, we compare MT-specific models and language models for (i) their **preference** when given an ambiguous subsentence, (ii) their **sensitivity** to disambiguating context, and (iii) the performance **disparity** between figurative and literal source sentences. We find that current MT models consistently translate English idioms literally, even when the context suggests a figurative interpretation. On the other hand, LMs are far more context-aware, although there remain disparities across target languages. Our findings underline the potential of LMs as a strong backbone for context-aware translation.

## 1 Introduction

Natural language is inherently ambiguous due to the competing pressures of efficiency and clarity in communication (Zipf, 1949; Piantadosi et al., 2012). As communicators, we disambiguate meanings on the basis of a wide range of contextual factors, or ask clarifying questions when such context is not available. Though sometimes overlooked, the role of ambiguity in NLP has gained growing interest in recent work (Min et al., 2020; Liu et al., 2023; Stengel-Eskin et al., 2023).

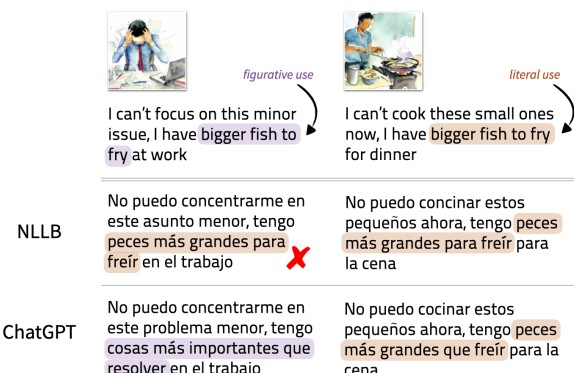

Figure 1: TIDE consists of pairs of contrastive sentences that contain the same idiomatic expression in different contexts, such that one uses the figurative meaning of the idiom (left), and another uses its literal meaning (right). On this set of inputs, ChatGPT is sensitive to the disambiguating context when translating the idiom, while NLLB is not.

In machine translation (MT), it has long been recognized that ambiguities arise when the source language does not encode grammatical attributes that the target language requires (Bar-Hillel, 1953; Prates et al., 2019; Savoldi et al., 2021; Gonen and Webster, 2020, i.a.). For instance, the English sentence "*I am a doctor*" would require disambiguating the doctor's gender for translation to German, which has no gender-neutral word for "*doctor.*" Prior work created contrastive test sets for such phenomena, to evaluate whether MT models correctly translate an ambiguous word (here, "*doctor*") when disambiguating context is available (e.g., "*She is a doctor*") (Müller et al., 2018; Bawden et al., 2018; Voita et al., 2019b).

In contrast with *grammatical ambiguity* with respect to a target language, it is relatively less understood how MT systems handle *semantic ambiguity* present in the source text itself. For instance, "*I have bigger fish to fry*" is ambiguous between figurative ("*... at work*") and literal ("*... for the dinner*") interpretations in English, outside of the

---

[1]Data and code can be found at https://github.com/jaechan-repo/mt-ambiguity.

context of translation. Therefore, we extend the line of work on context-aware translation to semantically ambiguous phrases in English.

To this end, we create TIDE, **T**ranslations of **I**dioms in **D**isambiguating context in **E**nglish, a dataset of 512 example triples. Each triple consists of an ambiguous subsentence and a pair of contrastive sentences that contain the subsentence but add disambiguating context: one to produce a figurative interpretation of the idiom, and another to produce a literal interpretation of it (see Figure 1 for an example). Our creation process for the triples combines automatic text generation with human annotation: we use GPT-4 to draft the triples, which are then scrutinized by human annotators. Following this, we engage native speakers from four languages to craft reference translations for a subset of the dataset.

In our experiments, we evaluate both traditional neural MT models and language models (LMs). MT-specific models are trained on large corpora of parallel sentences, and have formed the foundation of translation research; LMs are trained without any explicit supervision for translation, yet recently demonstrate impressive translation ability (Hendy et al., 2023). Using TIDE, we compare how these two types of systems handle ambiguity, and evaluate their sensitivity to disambiguating context. We find that on ambiguous input, LMs demonstrate roughly balanced preference between literal and figurative interpretations, whereas MT-specific models consistently prefer literal ones (§4.1). Given disambiguating context, LMs are substantially more context-aware, though this sensitivity declines for more low-resource target languages; in contrast, MT-specific models tend to translate idioms literally irrespective of context (§4.2). Finally, MT-specific models are better at translation of literal text than figurative text, whereas this disparity in LMs is much narrower (§4.3).

We summarize our contributions as follows: (1) We formalize the challenge of ambiguous idiomatic language in MT; (2) we create a new translation benchmark, TIDE, that includes sentences with idioms along with disambiguating contexts (literal and figurative); (3) we analyze MT systems' behavior with and without disambiguating contexts, pointing to interesting trends and differences between LMs and MT-specific models.

## 2 Creating TIDE

Idioms, though commonplace in daily communication, pose a challenge for MT systems due to its inherent ambiguity between literal and non-literal meanings. Generating the most appropriate translation among potential disambiguations of the idiom involves an understanding that extends beyond the idiom itself, as an MT system must use broader context clues to discern the most fitting translation.

We present TIDE, a dataset of 512 example triples. Each triple consists of an *ambiguous subsentence*, a *figurative sentence*, and a *literal sentence* in English, all including the same idiom. The ambiguous subsentence permits both figurative and literal interpretations of the idiom, while the figurative and literal sentences introduce additional context that resolves the ambiguity to figurative and literal readings, respectively. We design subsentences (e.g., *"had a card up his sleeve"*) to be more than an idiom itself (here, *"card up sleeve"*), as idioms alone can often be unnatural as standalone input to an MT system.

We construct TIDE through a human-AI collaborative approach following a line of recent work (Liu et al., 2022; Chakrabarty et al., 2022). We first manually select candidate idioms from two large idiom corpora (§2.1). Next, we leverage the generative power of GPT-4 to efficiently produce diverse and high-quality text, by prompting it to write a complete triple for each idiom (§2.2). To ensure quality and correctness, we then involve human annotators to filter out invalid triples (§2.3). Finally, we collect gold translations for a subset of the dataset among native speakers (§2.4).

### 2.1 Collection of Idioms

To collect idioms, we scrape THE IDIOMS dictionary[2] to obtain 1409 idioms, and additionally use a dataset of 905 idioms from Rabinovich et al. (2020); both sources contain corresponding idiom definitions. We discard duplicate idioms (including those that appear in different conjugations) and proverbs (e.g., *All that glitters is not gold*), which are often too self-contained to be disambiguated with context. Then, we manually select idioms that are available to a natural and plausible *literal* interpretation, in addition to their figurative meanings. This results in a set of 700 idioms with definitions.

---

[2] https://www.theidioms.com/

| Idiom | Figurative Sentence | Literal Sentence |
|---|---|---|
| tip of the iceberg
*to only know a very small part of the problem* | The problems we discovered were **just the tip of the iceberg** in this company. | As we approached the glacier, we saw **just the tip of the iceberg** above the water. |
| fall between the cracks
*be ignored or unobserved* | His request for a promotion **fell between the cracks** due to the company's restructuring. | The small toy **fell between the cracks** of the wooden floor. |
| foam at the mouth
*be extremely angry* | He **was foaming at the mouth** when he found out about the betrayal. | The rabid dog **was foaming at the mouth** and needed to be isolated. |
| foot in the door
*succeed with a first step* | By volunteering at the company, she **got a foot in the door** for a full-time position. | When the door was closing, he quickly **got a foot in the door** to prevent it from shutting. |

Table 1: **Examples in TIDE**. A figurative and literal sentence disambiguates the idiom by adding context that demands figurative and literal interpretations, respectively.

## 2.2 Generation of Idioms in Context

Next, we draft an example triple for each idiom by prompting GPT-4 with a fixed prompt, containing two in-context examples along with additional guidelines (details in Appendix A). We write a set of heuristics to automatically identify some types of ill-formed output, such as when the subsentence is not an exact substring of the full sentences. When a rule is violated, we add an additional turn of dialogue instructing the model to revise its output to follow the broken rule. We repeat this until all rules are followed, or when two revisions are attempted without success. After this, we have 700 English triples, each associated with a unique idiom.

## 2.3 Human Annotation

Of course, the triples collected in §2.2 may not correctly use idioms literally and figuratively, and generated text is susceptible to fluency and coherence issues. To ensure data quality, we recruit crowdworkers on Amazon Mechanical Turk to label each of the full sentences as using either the literal or the figurative sense of an idiom. We present each full sentence independently (not as a pair) to two different crowdworkers, who are asked to label it as *figurative*, *literal*, or *ambiguous* with respect to how it uses the given idiom. They may also indicate that the sentence is invalid if it is offensive or has fluency issues (see Appendix B for details).

The annotators achieved substantial agreement on this task, with a Fleiss $\kappa$ score of 0.721. Furthermore, for 82.9% of examples, there is a complete agreement between both annotators and the intended label (the label which we ask GPT-4 to follow when generating triples).

Based on the annotations, we discard triples where the intended-figurative sentences received no votes for figurative, or the intended-literal sentences received at least one vote not for literal. This asymmetry in the filtering heuristic is because we observe that GPT-4 was far more reliable at generating figurative uses of idioms than literal ones, and therefore we enforce a lower bar for retaining figurative sentences. We also discard all the triples that contain at least one vote for discard. In this way, we obtain the 512 English triples which constitute TIDE.

## 2.4 Collecting Translations

Finally, for a randomly subset of 50 idioms, we gather reference translations for the contrastive pairs of figurative and literal sentences from native speakers of Hebrew, Yoruba, Korean, and Chinese.

## 3 Experimental Setup

In this section we outline the models (§3.1) and languages (§3.2) we evaluate, our automatic metrics (§3.3), and our setup for collecting human evaluations of generated translations (§3.4).

### 3.1 Models

We evaluate two classes of translation systems: MT-specific models and LMs. Here, the MT-specific models use an encoder-decoder architecture and are trained on large amounts of parallel data, whereas the LMs are decoder-only models trained to maximize likelihood (i.e., next-token prediction) on predominantly-English text.

**MT-Specific Models** We evaluate NLLB (Meta, 2022) and Opus MT (Tiedemann and Thottingal, 2020; Tiedemann, 2020). NLLB is trained on par-

tially synthetic parallel data, and covers 202 languages.[3] `Opus MT` is a collection of models, each with a fixed source and target language.[4] For both models, we decode the translation greedily.

**Language Models** We evaluate `ChatGPT` (`gpt-3.5-turbo`; OpenAI 2022)[5] and `PaLM 2` (`text-bison-001`; Google et al. 2023).[6] We do not include `GPT-4` as it partially authored the examples in the dataset.

Both models were trained on a mixture of different languages, and in particular `PaLM 2`'s training corpus included parallel data for hundreds of languages. However, both LMs are trained for the next-token-prediction objective.

We prompt the LM to generate translations zero-shot with the prompt "`Translate the following English sentence to [target language]: [source sentence]`," and greedily decode the continuation. We do not provide in-context examples or further instructions about figurative language, in order to create a setting comparable to the evaluation of MT-specific models.

**Google Translate** We also include Google Translate[7] for reference due to its popularity in commercial use. We do not classify it as either an MT-specific model or LM due to the lack of public understanding of how it works.

## 3.2 Languages

We consider the eight target languages: Spanish (`Es`), Hindi (`Hi`), German (`De`), Hungarian (`Hu`), Korean (`Ko`), Chinese (`Zh`), Hebrew (`He`), and Yoruba (`Yo`), which vary in resource-availability and are typologically and culturally diverse. When the evaluation requires a gold translation, we focus on the last four languages for which TIDE contains human-written references.

---

[3] https://huggingface.co/facebook/nllb-200-3.3B

[4] The most recent model for each language pair was downloaded from https://github.com/Helsinki-NLP/Tatoeba-Challenge/tree/master/models: transformer-big for `De`, `Es`, and `Hu`, transformer-align for `He`, `Hi`, and `Yo`. Their most recent English to Chinese models by July 2023 do not produce coherent outputs, so we proceed with the earlier version available on HuggingFace: https://huggingface.co/Helsinki-NLP/opus-mt-en-zh. English to Korean models are not evaluated due to an issue with their PyTorch implementation, as reported by multiple users.

[5] API last accessed on June 18, 2023.

[6] API last accessed on June 17, 2023.

[7] https://translate.google.com/. API last accessed on June 14, 2023.

## 3.3 Automatic Metrics

We use different sets of metrics to evaluate translations for their literalness and for the overall translation quality.

**Literalness** Following Hendy et al. (2023), we use two metrics to assess the literalness of the translation: (1) *Unaligned Source Words* (USW) represents the number of source words unaligned with words in the translation, and (2) *Non-Monotonicity* (NM; Schioppa et al., 2021) determines the extent of reordering in the word-to-word alignments from the source sentence to its translation. For both metrics, we use the bitext alignments from the `awesome-align` framework (Dou and Neubig, 2021) which extract word alignments from `mBERT` embeddings.

**Translation quality** We evaluate translation quality based on sentence similarity between reference and predicted translations. We use `chrF` (Popović, 2015), `BERTScore` (Sun et al., 2022), and `BLEURT` (Sellam et al., 2020). `chrF` measures precision, recall, and F-score of character $n$-grams. `BERTScore` is a contextual embedding-based evaluation metric that leverages the pretrained language model.[8] `BLEURT` is a learned regression metric for automatic evaluation of generated text, which utilizes `BERT` for training on pairwise comparisons of reference and candidate sentences, calibrated on human quality judgments.

## 3.4 Human Evaluation

Due to the documented limitations of automatic evaluation for translation (Kasai et al., 2022), we additionally perform human evaluation of model-generated translations for Chinese, Korean, Hebrew, and Yoruba. We recruit one native speaker for each language, who are presented with the source sentences in each triple, along with generated translations from `NLLB`, `Opus MT`, `ChatGPT`, and `PaLM 2`. The model-generated translations are presented in a random order not shown to the annotator. For each sentence, they are asked: (1) Does the translation use the figurative meaning of the idiom, the literal meaning of the idiom, preserve the ambiguity due to an equivalent idiom in their language, or is it too nonsensical to determine? (2) Overall, is the translation perfectly correct, containing slight errors, or containing major errors? We use the same subset

---

[8] We use `XLM-RoBERTa-base` embeddings for `BERTScore`. (Conneau et al., 2020)

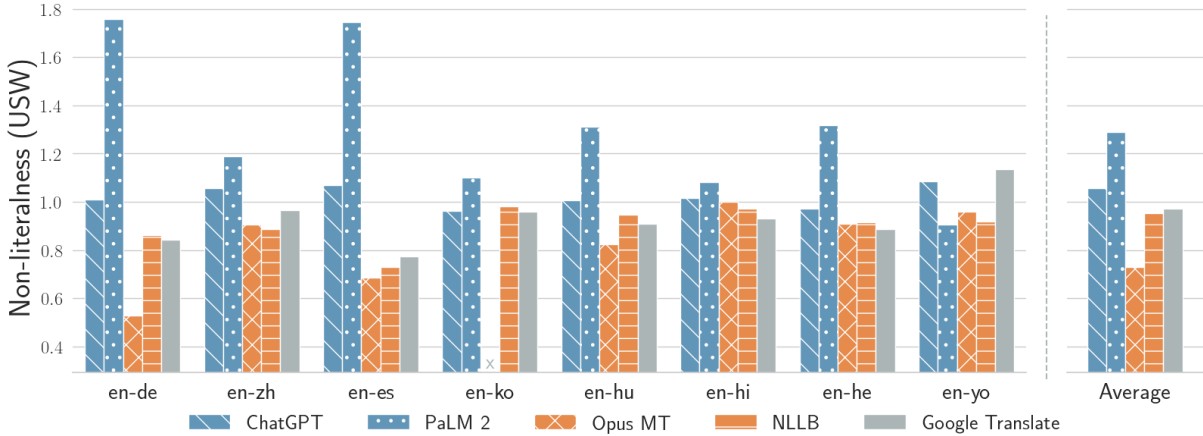

Figure 2: **Non-Literalness of Translations of Ambiguous Subsentences**, as measured by the number of *unaligned source words* (USW) between the source sentence and its translation, normalized by the within-language average. Translations from pretrained LMs are less literal than those of MT-specific models, suggesting that LMs prefer less literal translations of ambiguous input (i.e., without disambiguating context). En → Ko Opus MT models are not evaluated due to an issue with their implementation.

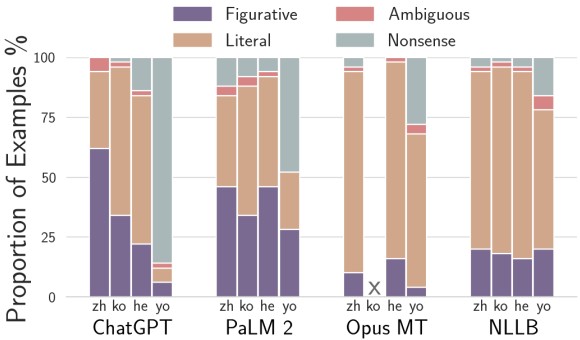

Figure 3: **Human Evaluation of Translations of Ambiguous Subsentences**, where annotators are asked to evaluate whether each translation is figurative, literal, ambiguous due to an equivalent idiom, or is nonsensical. ChatGPT and PaLM 2 are more balanced in their preference between figurative and literal translations; Opus and NLLB overwhelmingly prefer literal translations.

of 50 triples from §2.4. With 3 sentences per triple and 4 source models for each triple, annotators each evaluate 600 translations.

## 4 Experimental Results

In our experiments, we explore MT-specific and LM systems' translation behavior on ambiguous subsentences (§4.1), their sensitivity to disambiguating context (§4.2), and their overall competence at translating literal versus figurative input (§4.3).

### 4.1 RQ1: How do MT systems translate ambiguous subsentences?

First, we investigate how MT systems behave on ambiguous subsentences *without* disambiguating context, in order to measure their preference for translating them figuratively or literally. We hypothesize that LMs are more likely to produce less literal translations of ambiguous subsentences than MT-specific systems, based on recent findings in Raunak et al. (2023). Unlike their setting, here the source sentences are always ambiguous, so both literal and figurative translations are correct.

**Automatic Evaluation** We measure the literalness of translations using USW and NM, where higher values mean less literal translations (§3.3). Within each language, we normalize values by the average across systems in that language. This is because the metrics are not comparable across target languages, as they depend on linguistic properties of each target language. Shown in Figure 2, LMs (in blue) produce translations with higher USW scores than MT-specific models (in orange), across all target languages. In particular, Opus MT is the most literal model across all target languages. Moreover, we observe that the differences between LMs and MT-specific models become less pronounced for more under-resourced languages (the languages are ordered left to right based on count of pages in Common Crawl[9]).

---

[9] https://commoncrawl.github.io/cc-crawl-statistics/plots/languages

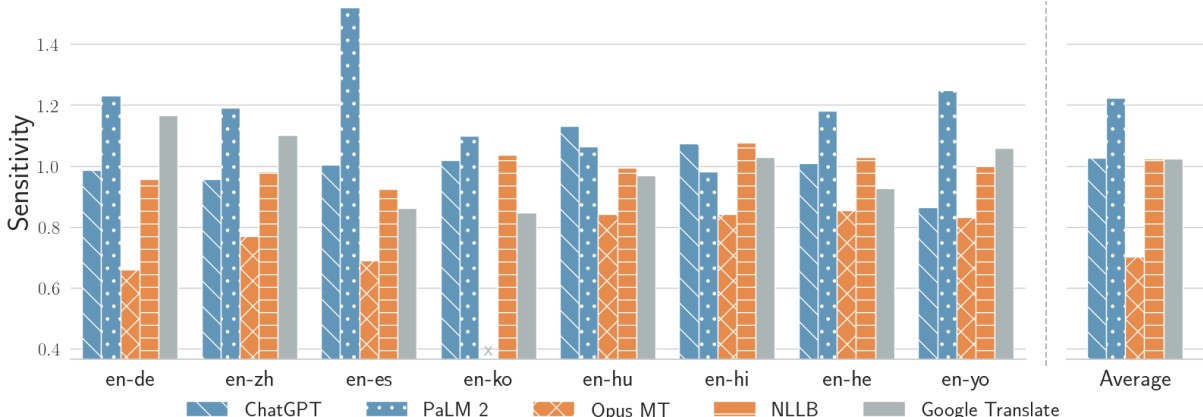

Figure 4: **Sensitivity to Disambiguating Context**, as measured by BERTScore-P, describes how well an MT model adapts to disambiguating context for an otherwise ambiguous subsentence. The metric is based on how the translation of the ambiguous subsentence changes between the two full sentences, and is normalized by the in-language mean. LMs generally demonstrate greater context-awareness than MT-specific models.

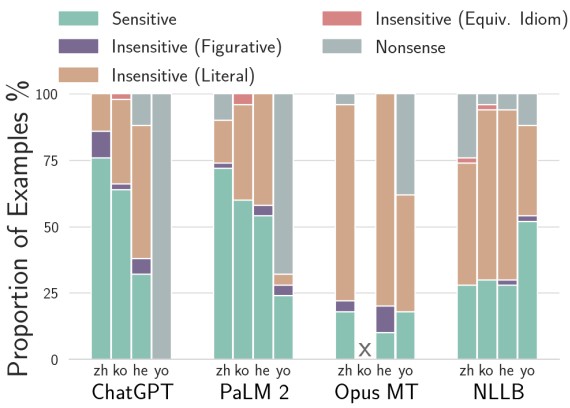

Figure 5: **Human Evaluation of Sensitivity**. A set of generated translations is considered context-sensitive when it uses the figurative (or literal) sense of an idiom given figurative (or literal) disambiguating context. ChatGPT and PaLM 2 are much more context-sensitive than Opus MT and NLLB, which tend to translate idioms literally irrespective of context.

Results based on NM (shown in Appendix C) corroborate our findings for SVO languages. This metric is inherently limited to target languages with the same word order as the source language (English in this work, with SVO order).

**Human Evaluation** In Figure 3, we show the human judgments of translations of ambiguous subsentences, indicating whether the translation is ambiguous, literal, figurative or nonsense. These results corroborate findings from automatic evaluation, and show even clearer distinctions. Overall, ChatGPT and PaLM 2 demonstrate much more balanced preferences between figurative and literal translations, compared to Opus MT and NLLB. For

the target language Chinese, ChatGPT prefers a figurative translation 62% of the time; however, that preference declines dramatically as the target language becomes more low-resource, dropping to 6% for Yoruba. PaLM 2 demonstrates more robust preferences across target languages, consistently preferring figurative translations 28% to 46% of the time. In contrast, Opus MT and NLLB overwhelmingly prefer literal translations, choosing a figurative translation only 4% to 20% of the time.

## 4.2 RQ2: How sensitive are MT systems to disambiguating context?

We next explore to what extent the predicted translation of an ambiguous subsentence changes when disambiguating context is available.

**Automatic Evaluation** Intuitively, if the LM is not sensitive to context, then the translation of the ambiguous subsentence, $p_a$, should be equally contained in the translation $p_\ell$ for the literal sentence, and the translation $p_f$ for the figurative sentence. That is, the way the ambiguous subsentence $a$ is translated should not be affected by the added context. On the other hand, if $p_a$ is more contained in $p_\ell$ than in $p_f$ (or vice versa), that would mean how the model handles $a$ changes with the context.

Therefore, we operationalize the sensitivity to disambiguating context as

$$|\texttt{contained\_in}(p_a, p_l) - \texttt{contained\_in}(p_a, p_f)|$$

where $\texttt{contained\_in}()$ is a measure of unidirectional sentence similarity. Here, we use chrP and

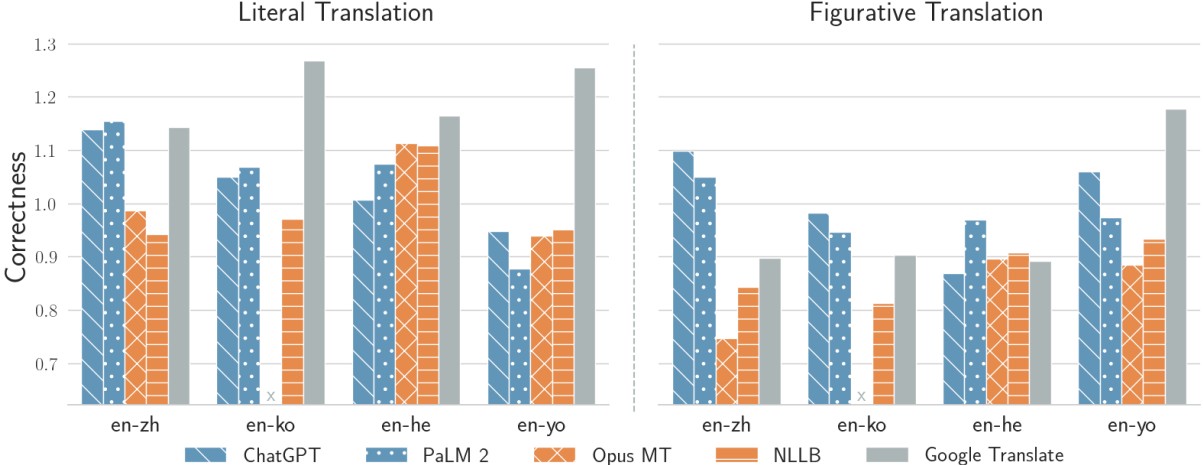

Figure 6: **Overall Translation Quality for Literal (left) and Figurative (right) Sentences**, as measured by BLEURT between the reference and prediction. While LMs and MT-specific models show comparable performance in translating literal sentences, NMT models are much weaker on figurative source sentences.

BERTScore-P, the precision outputs of chrF and BERTScore, both ranging from 0 to 1. A higher value of sensitivity (close to 1) indicates high sensitivity to disambiguating contexts.

Figure 4 shows the sensitivity results for the different models. The LMs, PaLM 2 and ChatGPT, generally exhibit a higher degree of sensitivity across most language pairs. Comparatively, the MT-specific models, Opus MT and NLLB, show less sensitivity. Opus MT, in particular, consistently demonstrates the lowest context sensitivity for all target languages.

**Human Evaluation** In human evaluation, a model is considered context-sensitive on a triple if annotators indicate that the idiom is translated figuratively for the figurative sentence, and literally for the literal sentence. Otherwise, the model is insensitive. As shown in Figure 5, both ChatGPT and PaLM 2 are very sensitive to context, though there is still room for improvement. For instance, for En→Zh translation, ChatGPT and PaLM 2 translate correctly for 76% and 72% of idioms, respectively. Yet, the sensitivity of both models declines monotonically as the target language becomes more low-resource. In particular, for En→Yo translation, ChatGPT translations are entirely nonsensical, and are qualitatively reported as frequently containing hallucinations completely unrelated to the source.

Nonetheless, Opus MT and NLLB are substantially less context-aware, correctly adapting to disambiguating context only 11.5% and 34.5% of the time, respectively. Yet, their more consistent performance across languages suggests that dedicated

training for translation leads to better results on low-resource languages.

### 4.3 RQ3: Are there performance disparities between figurative and literal translations?

Finally, we investigate if translation systems have systematic performance gaps between translating figurative versus literal input.

**Automatic Evaluation** We use the reference translations collected in §2.4, and measure text similarity between predicted and reference translation with BLEURT.

The results are shown in Figure 6. Across the board, models are more capable at literal translation than figurative translation. Yet, the gap is more pronounced for MT-specific models compared to LMs. ChatGPT and PaLM 2 exhibit performance gaps of 2.92% and 4.85%, respectively, between literal (higher) and figurative translations, on average across languages. For OPUS and NLLB this disparity is higher: 16.4% and 11.7%, respectively.

Overall, MT-specific models and LMs demonstrate comparable performance on literal translations, while NMT models lag behind LMs on figurative translations.

**Human Evaluation** In Figure 7, we compare how human annotators evaluate the correctness of translations overall, with the options *perfect*, *minor mistakes*, and *major mistakes*. Consistent with findings from automatic evaluation, ChatGPT and PaLM 2 demonstrate more consistent performance across

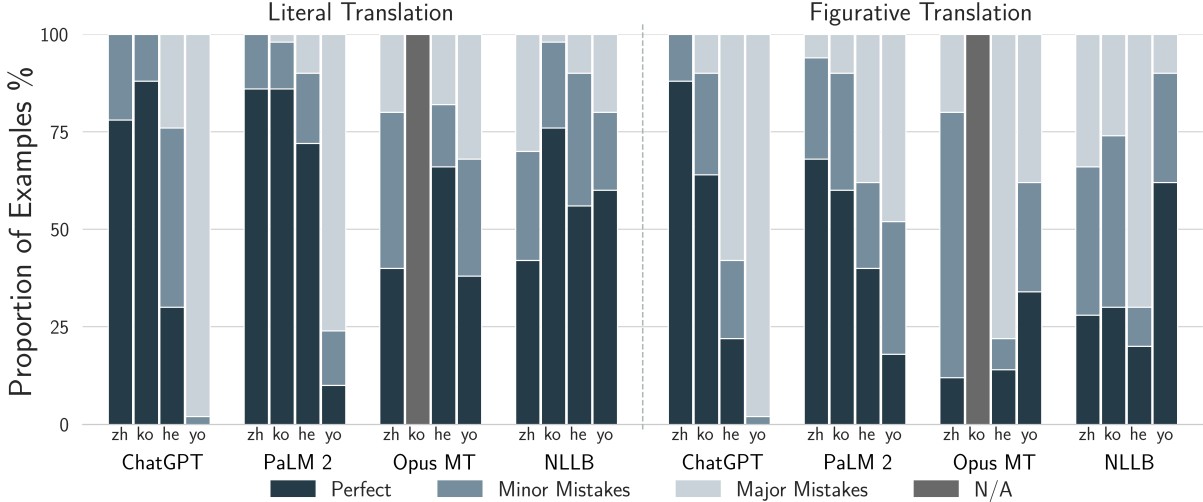

Figure 7: **Human Evaluation of Overall Translation Quality**, reported separately for figurative versus literal source sentences. `Opus` and `NLLB` are substantially better at literal translation than figurative translation overall, whereas `ChatGPT` and `PaLM 2` exhibit a much smaller disparity between literal and figurative translation quality.

literal and figurative translations. However, `Opus` and `NLLB` are notably stronger at literal translations than figurative ones.

We additionally observe that on Yoruba, the most low-resource language we study, `Opus MT` and `NLLB` actually far outperform `ChatGPT` and `PaLM 2`. We speculate that pretrained LMs are particularly strong on languages that were well-represented during pretraining; when this is not the case, it may produce degenerate text by entirely failing to grasp the translation task.

## 5 Related Work

**Ambiguity in translation** Context-aware translation usually focuses on grammatical features that the source language does not encode but the target language requires, such as formality (e.g., Chinese has a formal and informal "*you*"; Voita et al., 2019a), gendered pronouns (e.g., French has a male and female "*it*"; Müller et al., 2018; Yin et al., 2021), verb form (e.g., Spanish has six verb forms for past tense; Fernandes et al., 2023), and ellipses (e.g., "*We all did*" in English cannot be translated to Russian without identifying the elided verb; Voita et al., 2019b). Another well-studied issue is lexical cohesion, where the same phrase in the source sentence (e.g., a named entity like "*Julia*") should be translated consistently each time (Wong and Kit, 2012; Kuang et al., 2018). In contrast, our work extends the study of context-aware translation to expressions which are ambiguous *in the source language alone*, focusing on idiomatic expressions.

TIDE joins a family of contrastive datasets that test model sensitivity to contextual information (Müller et al., 2018; Bawden et al., 2018; Voita et al., 2019b, i.a.).

**Translation of figurative language** Figurative language has received considerable attention in MT research. Some work has studied the hidden representations or attention patterns of MT-specific models when processing multi-word expressions (Rikters and Bojar, 2017; Garcia et al., 2021; Dankers et al., 2022), or proposed methods to improve translation of these expressions (Zaninello and Birch, 2020; Gamallo and Garcia, 2019). In particular, Baziotis et al. (2023) show that monolingual pretraining improves figurative translation, which may explain our finding that pretrained LMs generate less literal translations and are more sensitive to disambiguating context.

The most closely related work, Raunak et al. (2023), compare how LMs and MT-specific systems translate sentences with idiomatic expressions, and similarly find that LMs produce substantially less literal translations. We go further by evaluating how these models handle *ambiguous* input and their *sensitivity* to disambiguating context.

**Datasets for idiom translation** Fadaee et al. (2018) introduced the first extensive dataset for idiom translation, identifying data scarcity as one of core challenges in this domain. EPIE (Saxena and Paul, 2020) is a large-scale corpus with 25K potentially idiomatic expressions (PIEs), with rep-

resentation of both figurative and literal usages. MAGPIE (Haagsma et al., 2020) is a more expansive dataset of 50K samples that also contain genre labels. PECTI (Tang, 2022) curated a parallel English translation dataset of Chinese idioms. While these datasets offer a general-purpose testbed, the contrastive sentence pairs in TIDE enable finer-grained analysis, while the fluency of source sentences matches (if not exceeding) that of naturally-occurring datasets.

## 6 Conclusion

In this work we focus on semantic ambiguity in machine translation, specifically when using idiomatic language. We introduce a new benchmark (TIDE) of sentences that include idioms, along with disambiguating contexts (both literal and figurative). We then use TIDE to investigate the behavior of different translation systems on ambiguous input and their sensitivity to disambiguating context, uncovering new strengths of pretrained LMs compared to MT-specific models.

Our findings point to pretrained LMs as a promising backbone for translation systems, and we foresee a future that combines the strong language understanding of LMs with dedicated supervision for translation.

## Acknowledgments

We would like to thank the UW NLP community for valuable discussion of this work. We are grateful to Weijia Shi, Jiacheng (Gary) Liu, and Xiaochuang Han for their help in writing and evaluating Chinese translations, and Zhaofeng Wu for feedback on the draft and figures.

We thank the reviewers for their valuable feedback and suggestions, and OpenAI for offering access to their models through the API.

## Limitations

In this work we study ambiguous source sentences specifically through idioms that are available to both literal and figurative interpretations. While this allows us to efficiently collect a dataset and perform focused evaluation, ambiguity occurs in more diverse forms, and we encourage future work to collect more data in the form of TIDE. Contemporary work collects a dataset of ambiguous sentences (with direct disambiguations, rather than disambiguating context), and is a promising start (Liu et al., 2023).

In addition, we only study the behavior of translation systems when English is the source language, due to the availability of English idiom collections. Yet figurative expressions vary greatly across languages (Kabra et al., 2023), and our conclusions may not necessarily generalize to translation from other languages.

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

## A   TIDE Creation Details

### A.1   Sentence generation

We use `GPT-4` to generate the 700 triples consisting of an ambiguous subsentence, a figurative sentence, and a literal sentence. The configuration parameters were set as follows: `max_tokens=512`, `temperature=0`, and `top_p=1`. The prompt is shown in Table 2.

In addition, the generation process undergoes iterative refinements under a set of criteria, during which we prompt the `GPT-4` instance to rewrite the entire triple if the generation included any prohibited words: "*literally*", "*figuratively*", "*ambiguously*", "*physically*", "*metaphorically*", and "*because*". These words are observed to potentially degrade sentence quality, as they often prompt the `GPT-4` to merely provide working definitions of the idioms instead of generating novel context. We also ensure through these refinements that the ambiguous subsentence is indeed a substring of the figurative and literal sentences.

### A.2   Processing pronouns

As written in the generation prompt (Table 2), we ban `GPT-4` from including subjects in the ambiguous subsentence as we observe that `GPT-4` frequently uses personal pronouns which end up disambiguating the whole subsentence (e.g., *He is a chip off the old block* is not ambiguous due to the pronoun *he*). Following the generation stage, we conduct additional rule-based modifications to the sentences to facilitate the translation process for MT models. In cases where the ambiguous subsentence begins with a lexical verb, and both the literal and figurative sentences include interchangeable subjects preceding the verb, we make alterations so that both use the same pronoun, which are then incorporated into the shared subsentence. These alterations include converting "he" to "she", "she" to "he", and "he"/"she" to "they" to have the pronoun shared between the figurative and literal sentence.

Figure 8: **Amazon Mechanical Turk (MTurk) Worker Interface**, containing 3 example problems.

## B   Amazon Mechanical Turk (MTurk) details

We employ Amazon Mechanical Turk (MTurk), a crowdsourcing marketplace, to collect well-formed triples, composed of an idiom and corresponding ambiguous subsentence, figurative sentence, and literal sentence, generated by `GPT-4` as described in §2.2.

We select 30 workers based on their scores in a qualification test human intelligence task (HIT) that we administer. This test, which typically requires less than 30 minutes to complete, consists of 20 handcrafted problems in the exact format as the main HIT. Upon completion, workers receive a payment of $7.

In both the qualification test and the main task, each problem presents an English utterance derived from a randomly shuffled pool of 700 ambiguous subsentences, 700 figurative sentences, and 700 literal sentences. The problem also provides the corresponding idiom in use and its dictionary (figurative) definition. With this information, workers must ascertain whether the idiom was used in a *figurative*, *literal*, or *ambiguous* context, or if the utterance should be *discard*ed. The option to discard is included to eliminate nonsensical or offensive generations.

For the main task, we gather multiple gold labels for each problem to ensure accuracy and credibility. This means that multiple workers are assigned the same problem. We initially release a batch of first 50 problems of the pool, collecting 4 gold labels for each to examine interannotator agreement. Based on our observation that 2 gold labels are sufficient, we proceed to collect only 2 labels for the remaining batches. Workers are remunerated at a rate of $0.30 for every 5 problems completed, an interval expected to take 1 minute.

For how we utilize these labels, see §2.3.

## C  Additional results

See Figure 9 and Figure 10 for additional results on RQ1 and RQ2, respectively.

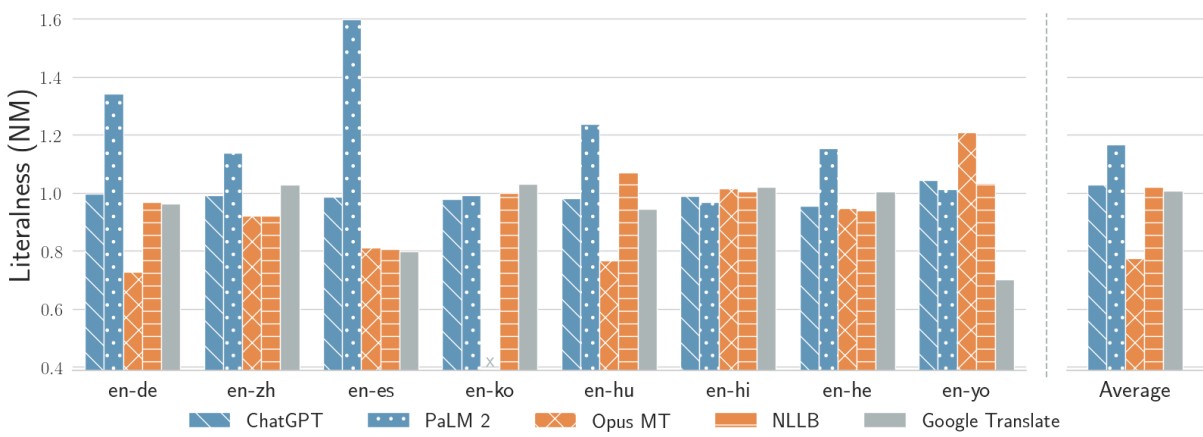

Figure 9: **Non-Literalness of Ambiguous Subsentences**. This graph visualizes non-monotonicity (NM) between the translation of each language pair, quantifying the level of word order shifts that occur during translation.

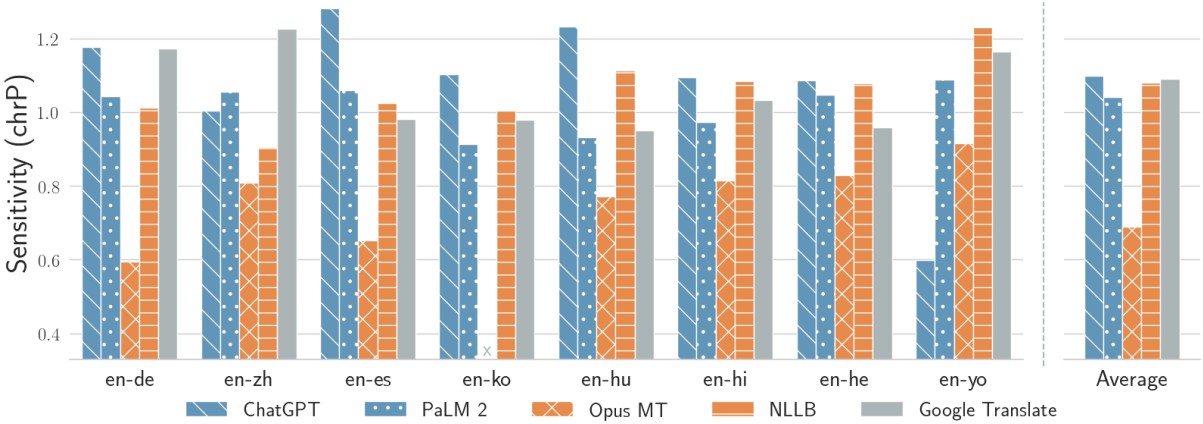

Figure 10: **Sensitivity**, $|\text{contained\_in}(p_a, p_f) - \text{contained\_in}(p_a, p_l)|$, computed using chrP (the precision score of chrF) as the implementation of contained_in.

**System Instruction**

System: Your task is to write an AMBIGUOUS phrase, a FIGURATIVE sentence, and a LITERAL sentence that use the given IDIOM. Consider the IDIOM "burn the bridge" as an example. An AMBIGUOUS phrase would allow for both figurative and literal interpretations of the IDIOM. For example, the phrase "burned the bridge with him" can be interpreted figuratively (in context of interpersonal relationships) or literally (burning down the physical bridge). A FIGURATIVE sentence must add extra words to your AMBIGUOUS phrase to only allow for a figurative, metaphorical interpretation. For example, the sentence "She burned the bridge with him because he publicly criticized her works" only makes sense in the figurative context of human relationships. A LITERAL sentence must add extra words to your AMBIGUOUS phrase to only allow for a literal, word-by-word interpretation. For example, the sentence "She burned the bridge with him after crossing it" only makes sense in literal context of a physical bridge that can be crossed. Note these warnings:

- Unless the idiom already contains a subject, your ambiguous phrase must not introduce any subjects.
- Do not end your ambiguous phrase with a preposition.
- Do not write more than one sentence for each figurative or literal sentence.
- Surround your AMBIGUOUS phrase with delimiters <>.
- When you write the LITERAL and FIGURATIVE sentences, do NOT modify the spelling of the AMBIGUOUS phrase in any way. The AMBIGUOUS phrase must be an exact substring of the FIGURATIVE and LITERAL sentences.

**In-Context Examples**

```
User: break the ice
```
```
Assistant: {
   "AMBIGUOUS": "<broke the ice with our teammates>",
   "FIGURATIVE": "we <broke the ice with our teammates> by initiating a conversation",
   "LITERAL": "holding steel ice hammers tightly, we <broke the ice with our teammates>"
}
```
```
User: felt pins and needles
```
```
Assistant: {
   "AMBIGUOUS": "<felt pins and needles>",
   "FIGURATIVE": "waiting for our boss to respond, we <felt pins and needles> in the room",
   "LITERAL": "she <felt pins and needles> on the floor when she spilled sewing supplies"
}
```

**Added Messages**

```
User: goose egg
```
```
Assistant: {
   "AMBIGUOUS": "<got a goose egg>",
   "FIGURATIVE": "he <scored a goose egg> after studying so diligently for the test",
   "LITERAL": "she <got a goose egg> from her grandfather's backyard"
}
```
```
User: The AMBIGUOUS phrase must be an exact substring of the FIGURATIVE sentence, but yours
isn't. Rewrite your AMBIGUOUS phrase, FIGURATIVE sentence, and LITERAL sentence accordingly.
```
```
Assistant: {
   "AMBIGUOUS": "<got a goose egg>",
   "FIGURATIVE": "he <got a goose egg> after studying so diligently for the test",
   "LITERAL": "she <got a goose egg> from her grandfather's backyard"
}
```

Table 2: **Prompt for GPT-4** to generate one triple. The instruction is stated once, followed by two complete in-context examples. Finally, only the idiom is provided for the last example. In this case, the generation does not meet the requirement that the ambiguous subsentence must be a substring of the figurative sentence. We request a new triple, to which GPT responds with a triple that meets all the requirements.