# OpenReview forum: "That was the last straw, we need more: Are Translation Systems Sensitive to Disambiguating Context?"
_EMNLP/2023/Conference — EMNLP 2023 Findings_

### Official Review · Reviewer_ZKwx · 2023-08-01

**Typos Grammar Style And Presentation Improvements:** 236-238
**Soundness:** 4

**Excitement:**

3: Ambivalent: It has merits (e.g., it reports state-of-the-art results, the idea is nice), but there are key weaknesses (e.g., it describes incremental work), and it can significantly benefit from another round of revision. However, I won't object to accepting it if my co-reviewers champion it.

**Missing References:**

The paper misses some relevant citations for new resources and/or analysis of MT and LMs for figurative language generation:

- Examining the Tip of the Iceberg: A Data Set for Idiom Translation (https://aclanthology.org/L18-1148, Fadaee et al., LREC 2018)
- MAGPIE: A Large Corpus of Potentially Idiomatic Expressions (https://aclanthology.org/2020.lrec-1.35, Haagsma et al., LREC 2020)
- Can Transformer be Too Compositional? Analysing Idiom Processing in Neural Machine Translation (https://aclanthology.org/2022.acl-long.252, Dankers et al., ACL 2022)
- Understanding Transformer Memorization Recall Through Idioms (https://aclanthology.org/2023.eacl-main.19, Haviv et al., EACL 2023)

The first two papers introduce larger datasets that were used for previous studies in idiom translation (like the third paper), and hence should be mentioned and compared to TIDE in the present work.

**Paper Topic And Main Contributions:**

This paper introduces TIDE, a new dataset to evaluate the figurative language capabilities of machine translation (MT) systems. TIDE was obtained via human-AI collaboration and contains 512 ambiguous idioms alongside disambiguating content for one possible figurative and literal usage. Authors use TIDE to analyze the generations of various MT systems and large language models (LLMs), focusing on 1) models' preferences between literal/figurative translations given ambiguous expressions; 2) models' sensitivity to disambiguating context; and 3) evaluating the translation quality gap between figurative and literal sources. The findings of this study corroborate the strong performance of LLMs in producing more context-aware translations compared to regular MT models while also showing the limited abilities of LLMs in handling low-resource settings.

**Questions For The Authors:**

A. Did you try to constrain GPT-4 to generate contrastive figurative/literal pairs that were minimally different from each other? While the current pairs may prove useful, a more granular analysis of the impact of context might be challenging due to their diversity.

**Reasons To Accept:**

- The paper is well-written and introduces a valuable resource for studying how figurative language is translated by machine translation systems and prompted LLMs.
- The experimental part is framed very clearly and involves both an automatic evaluation using appropriate metrics and a human evaluation covering a sizable data sample (when considering all languages and model combinations).
- Evaluation results are interesting and clearly showcase the strengths and weaknesses of each model category, covering a comprehensive set of state-of-the-art models and typologically-diverse target languages.

**Reasons To Reject:**

The main weakness of this work concerns the novelty of the collected data. Indeed, some ad-hoc test sets to evaluate idiom translation already exist (see "Missing References" section), although they were not mentioned in the Related Work section. It is unclear whether TIDE would prove more useful than the much larger MAGPIE dataset for studying figurative/literal translations with MT and LMs. For this reason, it would be important to discuss clearly which components make TIDE a relevant complement to existing resources.

**Reproducibility:**

3: Could reproduce the results with some difficulty. The settings of parameters are underspecified or subjectively determined; the training/evaluation data are not widely available.

**Reviewer Confidence:**

4: Quite sure. I tried to check the important points carefully. It's unlikely, though conceivable, that I missed something that should affect my ratings.

---

> ### Author Rebuttal · Authors · 2023-08-29
>
> Thank you for your insightful feedback on our submission! We are pleased that you recognize the soundness of our evaluation and “interesting” results that “showcase the strengths and weaknesses of each model category.”
>
> **Previous work - Need for a new dataset:** For a controlled investigation into model sensitivity to context, we need contrastive pairs with shared subsentences to compare translations before and after disambiguation. Many existing datasets based on naturally occurring examples, including MAGPIE, do not meet this constraint. Thanks to the reviewer for the references; we will include all of these in the next revision.
>
> **Q.A. Constrain GPT-4 to generate contrastive figurative/literal pairs that were minimally different from each other:** The reviewer’s suggestion to constrain GPT-4 with minimal pairs provides an interesting angle for creating this dataset. From a preliminary investigation following the reviewer’s suggestion, however, we found that efforts to collect minimal pairs led to the model generating subpar contexts (i.e., less contrastive than without the minimality constraint). Thus, it is not feasible to obtain minimally different pairs for most idioms (getting pairs is not always possible even without the minimality constraint). Such pairs of sentences also sound more unnatural and adversarial, rather than reflecting more natural uses of idioms. Moreover, lacking sensitivity even with non-minimal pairs of contexts is a more meaningful finding. This means that despite the extra surface-level differences (that minimal pairs lack) like length, word choices, or syntax, the model still fails to disambiguate the subsentence correctly.
>
> **Reproducibility:** We make the evaluation code and dataset publicly available to enhance the replicability of our study.
>
> Thank you for suggesting **presentation improvements** and pointing out the **missing references**, all of which we find constructive and immediately actionable. We will make sure to add them to the final version.

---

### Official Review · Reviewer_PHaX · 2023-08-04

**Soundness:** 3

**Ethical Concerns:**

Yes

**Excitement:**

4: Strong: This paper deepens the understanding of some phenomenon or lowers the barriers to an existing research direction.

**Justification For Ethical Concerns:**

As far as I've understood use of amazon mturk especially with annotators of less-resourced language experts should be justified. There is no indication of this in the paper and no section.

**Paper Topic And Main Contributions:**

Article describes testing of neural machine translations of idiomatic phrases in both non-idiomatic and idiomatic contexts. This is an interesting experiment and something that probably is both important for users of translation systems as well as problems for many human translators as well. The idioms and their translations are evaluated on mt-specific and generic large language models. The idiom translations are evaluated based on being literal, figurative or preserving ambiguity across languages.

**Questions For The Authors:**

A) What is the baseline quality of the used translation systems in easier sentences? I am thinking if English-Yoruba generates non-sensible tranlsations altogether, evaluating on the harder task of idiom translation might not be very effective use of resources.

**Reasons To Accept:**

* new testing approach for evaluation of idiom translations is interesting for most conference-goers in this venue

**Reasons To Reject:**

* very specific topic and potentially premature for the smaller languages

**Reproducibility:**

3: Could reproduce the results with some difficulty. The settings of parameters are underspecified or subjectively determined; the training/evaluation data are not widely available.

**Reviewer Confidence:**

4: Quite sure. I tried to check the important points carefully. It's unlikely, though conceivable, that I missed something that should affect my ratings.

---

> ### Author Rebuttal · Authors · 2023-08-29
>
> Thank you for your insightful feedback on our submission! We are pleased that you recognize “our new approach for evaluation” and that handling figurative language is “important for users of translation systems.”
>
> **Translation of figurative language has universal implications:** Figurative language is everywhere in real-world language use, and for broad application, strong MT models must correctly handle figurative language by attending to the disambiguating context. Aside from creating a diverse benchmark for future work on this topic, we also draw conclusions about the qualitative differences between large pre-trained LMs versus dedicated MT models for translations. The reviewer’s criticism that the work is too niche (“very specific topic and potentially premature for the smaller languages”) is indeed one of the heuristics discouraged by ACL 2023 peer review policies ( https://2023.aclweb.org/blog/review-acl23/#2-check-for-lazy-thinking ).
>
> **Q.A - Translations are generally sensible:** As shown in Figure 7, all models demonstrate commendable translation performance. The only exceptions are ChatGPT and PaLM 2 when handling Yoruba; however, explicitly supervised models such as Opus MT and NLLB exhibit satisfactory performance on translations of Yoruba as well. A native speaker who reviewed their Yoruba translations found them generally reasonable, particularly for the literal sentences.
>
> **Ethical Concerns - We did not use MTurk for data annotation in languages other than English:** Language experts for translation tasks were native speakers who volunteered to do the task (researchers in NLP). Only GPT-4 generation filtering (a fully English task) was completed by English-speaking MTurk annotators from the United States.
>
> **Reproducibility:** We make the evaluation code and dataset publicly available to enhance the replicability of our study.
>
> Your concerns have helped us identify areas for further clarification, and we appreciate your thoughtful critique!

---

### Official Review · Reviewer_eu4q · 2023-08-05

**Soundness:** 3

**Excitement:**

3: Ambivalent: It has merits (e.g., it reports state-of-the-art results, the idea is nice), but there are key weaknesses (e.g., it describes incremental work), and it can significantly benefit from another round of revision. However, I won't object to accepting it if my co-reviewers champion it.

**Paper Topic And Main Contributions:**

Summary: The paper is a comparison between LLM-based translation vs. the state of the art MT (NMT in this case) in handling translation of phrases that have both literal and figurative meanings.  The work includes semi-manual creation of a corpus of triples of ambiguous phrase along with its literal and figurative (idiom) usage.  Furthermore, authors conduct run the ambigious text through two LLM-based systems (openAI and PaLM) and also two NMT systems and assess the translation quality both automatically and through human judgement.  Result show that LM-based translation outperforms the NMT models, specially around figurative examples.

**Questions For The Authors:**

1. Don't you think using GPT to generate the literal and non-literal examples is a limit on your corpus?  Yes, there's a human review after that, but it seems that's just vetting and does not involve writing any example for those cases that GPT has failed to generate good examples.

2. Continuing with the above question, don't you think that gives an advantage to the LLM-based approaches?   A more comprehensive way of corpus creation would have been to compile authentic examples from a large corpus via search rather than using a generative model

**Reasons To Accept:**

1. MT for figurative text and idioms is fairly under-studied problem and this work is a step to close the gap.

2. The TIDES corpus can be a good benchmark for future studies on this topic.

**Reasons To Reject:**

1. The novelty of the work is quite limited.  It's suited for a focused workshop on MT or figurative languages rather than the main track of the tier-1 EMNLP.

2. The fact that an LLM has been used to create the examples of the TIDES corpus is a major limit and can even create a circular limit to the approach.  That limits the scope of examples to what GPT model is capable of generating and to some extent gives a leverage to the GPT-based solution.

3. The main approach is not explained well (some questions to follow)

4. Weak replicability: Due to the black-box nature of the study with the evolving GPT models, it is difficult to replicate these experiments.

**Reproducibility:**

3: Could reproduce the results with some difficulty. The settings of parameters are underspecified or subjectively determined; the training/evaluation data are not widely available.

**Reviewer Confidence:**

4: Quite sure. I tried to check the important points carefully. It's unlikely, though conceivable, that I missed something that should affect my ratings.

---

> ### Author Rebuttal · Authors · 2023-08-29
>
> Thank you for your insightful feedback on our submission! We are pleased that you recognized that TIDE can be a “good benchmark for future studies” and that we study a “fairly under-studied problem.” We hope the following discussion addresses your concerns.
>
> **Novelty - Translation of figurative language is an important topic:** Figurative language is everywhere in real-world language (consider our community’s recent series of Workshops on Processing Figurative Language, which the reviewer mentions), and for broad applicability, strong MT models must correctly handle figurative language by attending to the disambiguating context. Aside from creating a diverse benchmark for future work on this topic, we also draw conclusions about the qualitative differences between large pre-trained LMs versus dedicated MT models for translation. The reviewer's criticism of the novelty of our work based on the argument that our work is too niche ("better suited for a focused workshop on MT or figurative languages") is indeed one of the heuristics discouraged by ACL 2023 peer review policies.
>
> **Creating data using GPT (Q.1)**
> 1. **It is hard to collect contrastive pairs from a corpus of natural examples:** For a controlled investigation into model sensitivity to context, we need contrastive pairs with shared subsentences to compare translations before and after disambiguation. Our initial approach was to scrape figurative sentences from BookCorpus and theidioms.com, and then minimally revise them to create a literal counterpart. However, sentences from BookCorpus were often extracted from long dialogues/passages, scarcely understandable on their own. For sentences from theidioms.com, we found that minimal revisions were often insufficient for making the sentences literal, requiring the entire sentence to be changed. Thus we opted for the automatic generation of contrastive pairs.
> 2. **We tried crowdworker revision to improve model generations, but it was not helpful:** Indeed, we only removed incoherent LM generations instead of attempting to revise them. In fact we experimented with crowdworker revision, but found that it often failed to fix incoherence, mainly when the underlying issue was that the subsentence was not actually ambiguous. For instance, revising a phrase like “Playing soccer was a piece of cake” (erroneously marked as ambiguous by GPT-4) would require complete rewriting, as the idiom is already used figuratively without marked figurative context. We decided to remove it when complete rewriting is needed. We acknowledge that our corpus may exclude idioms that GPT-4 couldn’t handle, but we believe the decision maintains our dataset’s quality without affecting our findings.
> 3. **Human data creation from scratch suffers limited diversity:** Many studies have shown that crowdworkers employ heuristics to write examples efficiently, and resulting datasets suffer a lack of diversity ([Geva et al., 2019](https://aclanthology.org/D19-1107/); [Gururangan et al., 2018](https://aclanthology.org/N18-2017/)). In our case, crowdworkers may write short, prototypical uses of idiom, rather than longer and more natural-sounding sentences like the ones we observe in our dataset. Our work follows a growing body of work that points to the promise of using controlled text generation alongside human review for dataset creation ([Yuan et al., 2021](https://openreview.net/forum?id=Fkpr2RYDvI1); [Liu et al., 2022](https://aclanthology.org/2022.findings-emnlp.508/); [Bartolo et al., 2022](https://aclanthology.org/2022.naacl-main.275/); [Perez et al., 2022](https://aclanthology.org/2023.findings-acl.847/)).
>
> **Q.2 - Discussion of an unfair advantage to GPT-3.5:** We took care to use a different GPT model for generating dataset examples (GPT-4) than in our experiments (GPT-3.5), for which we also use PaLM2. We chose a more robust model for data generation than for evaluation, as we expect there are idioms that GPT-4 can handle, which GPT-3.5 cannot. (This would be consistent with other findings ([Liu et al., 2023](https://arxiv.org/abs/2304.14399)) that GPT-4 is substantially stronger than GPT-3.5 in dealing with ambiguity.) Additionally, translation is not part of the data creation pipeline, while it is the objective for evaluation.
>
> **Replicability of the use of GPT:** GPT models are indeed black box models. But for reasons outlined above, we follow a long line of work in using GPT-3/4 for data generation, and include GPT-3.5 in the set of models we evaluate because they are currently indispensable for characterizing the capacity of LMs for translation. Despite the challenges, we make the evaluation code and dataset publicly available to enhance the replicability of our study. We believe the value of the TIDE dataset for the research community is great.
>
> Your concerns have helped us identify areas for further clarification, and we appreciate your thoughtful critique!

---

### Meta-Review · Area_Chair_y2RJ · 2023-09-20

**Recommendation:** 4

**Metareview:**

This work presents a datsaet and evaluations for the ability of neural MT systems and language models to translate idiomatic or non-idiomatic text which is ambiguous alone but disambiguated with context. The reviewers had somewhat ambivalent but overall positive evaluations of the work’s soundness and interest to the field; moderate concerns were raised both with the distinctness of the dataset from prior work and the validation of the LLM-assisted generations that comprise the dataset.

The work merited discussion among the AC and reviewers, and it was agreed that the initial concerns of the reviewers regarding scope were not at issue, and that there remain some concerns as to related work and dataset composition, as stated above. Overall, however, I’m happy to see the community leveraging strong generative models to help ask interesting questions about model behavior. However, concerns as to whether a dataset co-generated by GPT-4 will lead to, e.g., ChatGPT performing better than had the data come from an unrelated distribution are serious and under-explored; see, e.g., the AlpacaEval results that show that strong generative models like GPT-4 and Claude2 prefer their own family of models at a greater rate (https://tatsu-lab.github.io/alpaca_eval/). I encourage the authors to engage with this.

---

### Decision · Program_Chairs · 2023-10-07

**Decision:**

Accept-Findings

**Comment:**

This work presents a datsaet and evaluations for the ability of neural MT systems and language models to translate idiomatic or non-idiomatic text which is ambiguous alone but disambiguated with context. The reviewers had somewhat ambivalent but overall positive evaluations of the work’s soundness and interest to the field; moderate concerns were raised both with the distinctness of the dataset from prior work and the validation of the LLM-assisted generations that comprise the dataset.

The work merited discussion among the AC and reviewers, and it was agreed that the initial concerns of the reviewers regarding scope were not at issue, and that there remain some concerns as to related work and dataset composition, as stated above. Overall, however, I’m happy to see the community leveraging strong generative models to help ask interesting questions about model behavior. However, concerns as to whether a dataset co-generated by GPT-4 will lead to, e.g., ChatGPT performing better than had the data come from an unrelated distribution are serious and under-explored; see, e.g., the AlpacaEval results that show that strong generative models like GPT-4 and Claude2 prefer their own family of models at a greater rate (https://tatsu-lab.github.io/alpaca_eval/). I encourage the authors to engage with this.